# Evaluating continuous nanosecond pulsed electric field (nsPEF) treatment as a non-thermal alternative for human milk pasteurisation

Yiting Wang[1], Farzan Zare[1,2], Elisabeth K. Prabawati[1], Buddhi Dayananda[1], Negareh Ghasemi[3], Firuz Zare[4], Paul N. Shaw[5], Pieter Koorts[6], Nidhi Bansal[1]*

1 School of Agriculture and Food Sustainability, The University of Queensland, St Lucia, Queensland, Australia, 2 School of Mechanical and Mining Engineering, The University of Queensland, St Lucia, Queensland, Australia, 3 School of Electrical Engineering and Computer Sciences, The University of Queensland, St Lucia, Queensland, Australia, 4 School of Electrical Engineering and Robotics, Queensland University of Technology, Brisbane, Queensland, Australia, 5 School of Pharmacy, The University of Queensland, Woolloongabba, Queensland, Australia, 6 Royal Brisbane and Women's Hospital, Herston, Queensland, Australia

* n.bansal@uq.edu.au

## Abstract

This study investigated the application of nanosecond pulsed electric field (nsPEF) treatment as an alternative pasteurisation method for donor human milk (DHM). A 0.1% saline solution was identified as the closest imitation to the received DHM in terms of pulse waveform and conductivity, which was used for the optimisation of PEF parameters. Complete inactivation of inoculated *Escherichia coli* was achieved after nsPEF treatment in saline with an initial count of 5 log CFU/mL and nearly a 7 log CFU/mL reduction with an 8 log CFU/mL initial count. In DHM, nsPEF treatment resulted in a 3 log CFU/mL reduction with an initial 5 log CFU/mL count and a 5 log CFU/mL reduction at higher initial counts. However, no statistically significant difference in log reduction was observed across various initial bacterial counts in DHM samples. Microscopic analysis revealed potential protective effects of human milk fat globules and epithelial cells on *E. coli*, resulting in residual counts of 2–3 log CFU/mL post-treatment. Overall, the maximum temperature during nsPEF treatment was approximately 36°C, highlighting its advantage over thermal pasteurisation, and further optimisation could be conducted to evaluate the potential protective effects of the milk components.

## Introduction

Fresh human milk contains an array of important immunoprotective and nutritional biomolecules. Unlike infant formula, maternal milk is considered nature's "complete food" for infants due to its high nutrient density and the presence of non-allergenic functional proteins, which provide innate immunity. Human milk is essential for facilitating neonatal growth and development for at least the first six months following birth

**Data availability statement:** All relevant data are within the paper and the repository from The University of Queensland eSpace (DOI: https://doi.org/10.48610/857a6ac).

**Funding:** Children's Hospital Foundation Innovator Grant (50317) and the National

Health and Medical Research Council Ideas Grant (GNT1182038).

**Competing interests:** The authors have declared that no competing interests exist.

[1]. It enhances infants' immunity and reduces the risk of necrotizing enterocolitis (NEC), gastroenteritis, respiratory infection, allergy, neurological disorders, cardiovascular disorders, and other chronic diseases in adolescence and adulthood [2–8]. Unfortunately, not all newborns can receive milk from their own mothers for various reasons. Consequently, donor human milk (DHM) is needed, and human milk banks are established. All received DHM must undergo a strict screening procedure and be pasteurised (in Australia) before it can be given to infants. However, the current pasteurisation technique, known as Holder pasteurisation (HoP) uses high-temperature processing that harms many vital HM proteins and enzymes and prevents HM from providing its advantages [9]. Therefore, a non-thermal pasteurisation method, such as Pulsed Electric Field (PEF) may serve as an ideal alternative to the conventional methods. In this technique, a high-voltage electric field (EF) is applied to foods placed between two electrodes in a treatment chamber for a short duration of time (usually ranging from 1 to 100 microseconds (µs)) [10,11]. Recently, nanosecond PEF (nsPEF) has been developed with a very short pulse duration, including a high EF of a few hundred kV/cm, delivering energy low enough to avoid the heating effect and hence, can be utilised as a non-thermal treatment [12,13].

Despite many encouraging results from PEF with different types of foods (e.g., bovine milk, juice) reported in numerous publications, only a few studies have explored PEF treatment with HM. A study by Sanchaya, Sujatha [14] compared raw DHM to PEF-treated DHM samples and reported effective microbial reduction with no significant changes in pH, acidity, or fat content. Similarly, Indumathi, Sujatha [15] applied PEF in DHM and also reported no significant change in pH, acidity, fat, lactose, and ascorbic acid content between PEF-treated and raw DHM. However, a significant reduction in protein was observed, which the author attributed to potential deposition of milk solids on the electrode surfaces. In our previous study, a 4 log CFU/mL reduction in total bacterial count was achieved through the PEF system using Response Surface Methodology, while better retaining bioactive components compared to the conventional heating method in a static system with an HM volume of 0.8 mL [16]. Therefore, this study aims to scale up the sample volume in a continuous PEF system and investigate the microbial safety of PEF-processed DHM. In semi-liquid and liquid products, continuous-flow treatment chambers, which are modified from static treatment chambers, are widely used as they allow the products to pass through with the aid of a circulating pump and have higher uniformity for bacterial inactivation [17]. However, the continuous system is more complicated than the static chambers since more parameters were involved in the processing. Thus, more restrictions and limits control the system than PEF processing in a static mode. Due to the preciousness of limited DHM, saline (NaCl) solution was first analyzed to imitate DHM samples at the matching pulse width, shape, and electric field strength. This also allows for investigating the role of the food matrix in bacterial inactivation. The saline solution with the matched characteristics and DHM was then inoculated with *Escherichia coli* at various concentrations and analyzed for microbial safety.

## Materials and methods

### Sample preparation

Sodium chloride (NaCl; ChemSupply, Australia) was dissolved in ultra-pure (Milli-Q) water to prepare saline solutions at various concentrations ranging from 0 to 0.2% (w/v). The solutions were sterilized using an autoclave at 121°C for 15 min and then stored in the refrigerator at 4°C until use. Donor human milk (DHM) was collected from donors by clinical staff at the Royal Brisbane Women's Hospital (RBWH, Queensland, Australia) in 2020. Before participation, a verbal explanation of the research project and procedure was given to the donors, and written informed consent was obtained. Ethics approval (project 2019002894) for this study was provided by the RBWH Human Research Ethics Committee (HREC) and the University of Queensland's Human Research Ethics Committee on 19 December 2019. To ensure donor confidentiality, all DHM samples were deidentified before being released to the researchers. The samples were stored in a freezer at −80°C until use. Before the experiment, the deidentified DHM samples were thawed overnight in the refrigerator (4°C) and pooled to a total volume of 1 L. The samples were then pasteurised using conventional heating methods, where sterile bottles containing DHM samples were heated in a water bath until the temperature reached 62.5°C and maintained for 30 min. After the pasteurisation, the milk samples were cooled to room temperature before bacterial inoculation. This pasteurisation step was carried out to eliminate the effect of natural microflora in the DHM to allow for the study of standardised inoculation.

The Gram-negative bacterial culture of *Escherichia coli* (JM109) was incubated for 24h to reach a concentration of 9 log colony-forming units (CFU) per mL. The bacteria were then washed twice by centrifuging with sterile 0.85% (w/v) saline solution at 10,621 x g for 5 min (Sigma 2-16P, Germany) and suspended in the sterile saline solution. This *E. coli* strain was chosen because it is commonly used for disinfection applications as a bacterial model and has low pathogenicity. The saline solutions and pasteurised DHM samples were inoculated with the washed *E. coli* suspension and diluted to achieve the target initial count (5, 7, or 8 log CFU/mL) in a final volume of 100 mL.

### Microbiological enumeration of the samples

The samples were serially diluted with sterile 0.85% (w/v) saline solutions. Tryptone Soya Agar (TSA), made from Tryptone Soya Broth (TSB) and bacteriological agar (OXOID, UK), was used to enumerate samples both before and after the PEF treatment using the pour plate method. The plates were incubated at 37°C for at least 48h before enumeration.

### Continuous PEF treatment

As presented in Fig 1 created with BioRender, a medium-energy pulsed power generator (MEPPG; MPC3010S-50LP, Suematsu Electronics Co., Ltd, Japan) was used to deliver energy up to 0.8 J per pulse (with a 500 Ω resistive load) and peak amplitude up to 30 kV. An infrared thermal camera (FLIR TG165, Teledyne FLIR, USA) was used to measure the temperature of the sample both before and after each pulsing step. Additionally, a fibre optic (FO) temperature sensor (FOTEMP-OEM 4-channel from Micronor Sensors with OPTOCON TS2 FO Temp) was used to monitor the transient temperature during the PEF treatment. Polycarbonate cuvettes with stainless steel electrodes (SS316) were custom-built by The University of Queensland's advanced manufacturing workshop to the desired dimensions (L x W x D: 40 mm x 2.65 mm x 1.65 mm). The flow rate was set at 100 mL per minute using a peristaltic pump (LABS3/UD15 Mini S Series Peristaltic Pump, Lab Direct, Australia). The sample reservoir, cuvettes, and pump were connected by a 5 mm internal diameter tubing of a total length of 120 cm (POPE Clear Vinyl tubing, Toro Australia Pty Ltd, Australia). A magnetic stirrer was placed under the sample reservoir together with a stir bar inside the reservoir during the PEF treatment to ensure sample uniformity. The sample reservoir was jacketed with an ice pack to aid cooling. Based on the previous studies with Milli-Q water [18], a total of 240,000 pulses (60,000 pulses per step, 5 min resting time between each step) were applied at 50 Hz frequency using a pulsed power generator as the optimised processing condition. As presented in Table 1, voltage pulses with 41–128 ns

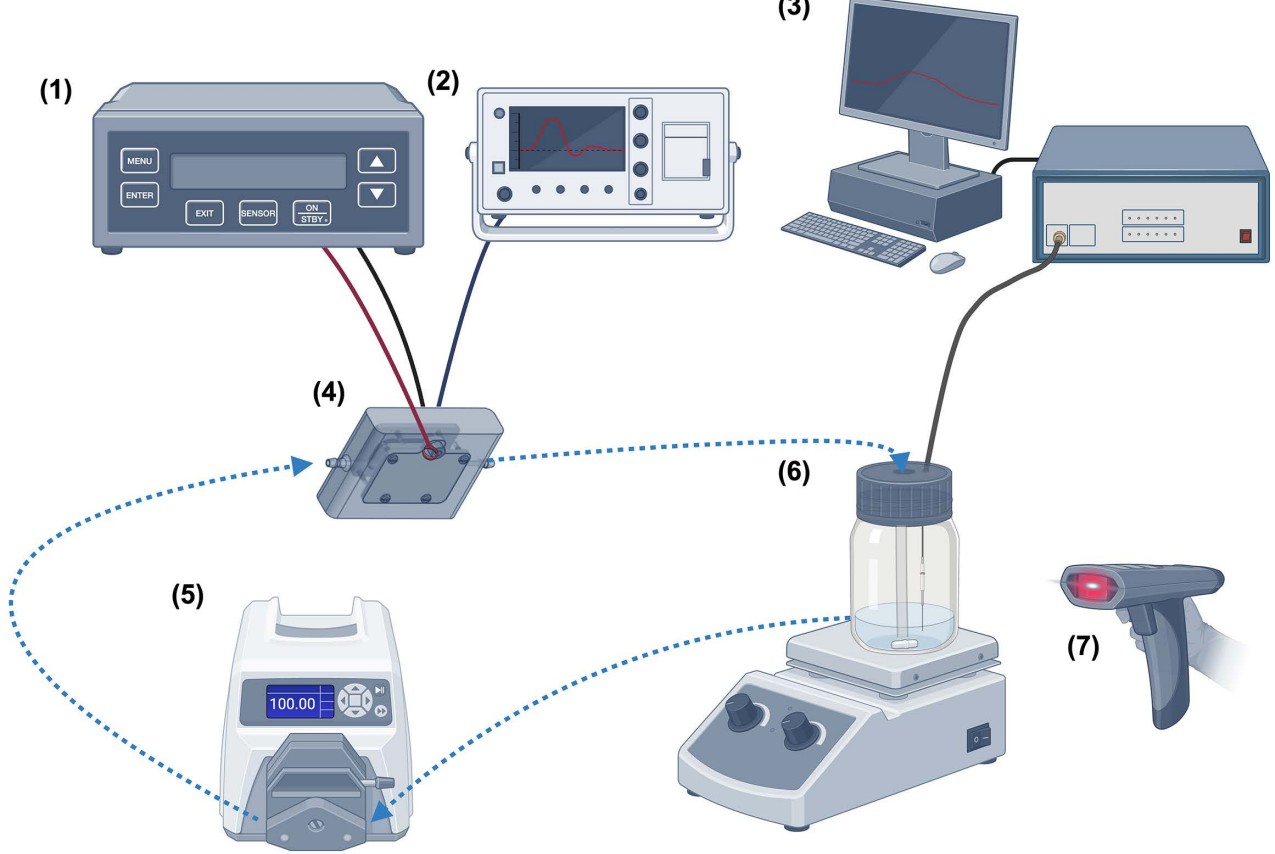

**Fig 1. Schematic diagram of the PEF system used in the study (1) Pulsed power generator, (2) Digital oscilloscope for voltage and current waveform, (3) Fiber optic temperature sensor and computer, (4) Cuvette, (5) Peristaltic pump, (6) Sample reservoir and magnetic stirrer plate, (7) Infrared thermal camera.** The blue dashed line represents the liquid flow. The liquid sample flows directly between the parallel plate electrodes in the cuvette and is not enclosed within the plastic tubing during exposure. Created with BioRender.com. Reproduction is based on a figure from [18].

**Table 1. Parameters of PEF processing in the continuous system.**

| Parameters | MQ | 0.05%S | 0.1%S | 0.2%S | HM |
|---|---|---|---|---|---|
| Media | Milli-Q water | 0.05% saline | 0.1% saline | 0.2% saline | Human milk |
| Peak Voltage (kV) | 21.8 | 20.3 | 19.3 | 17.9 | 19.2 |
| Peak current (A) | 52 | 100 | 194 | 254 | 148 |
| Electric field strength (kV/cm) | 93 | 123.0 | 117.0 | 108.5 | 116.4 |
| Pulse width (ns) | 2276 | 128 | 76 | 41 | 68 |
| Specific energy input (kJ/L) | 604.8 | 544.8 | 612.0 | 693.6 | 556.8 |
| Pulse per steps | 60,000 | 60,000 | 60,000 | 60,000 | 60,000 |
| Total pulse number | 240,000 | 240,000 | 240,000 | 240,000 | 240,000 |
| Frequency (Hz) | 50 | 50 | 50 | 50 | 50 |
| Flow rate (mL/min) | 100 | 100 | 100 | 100 | 100 |

pulse width were applied in the saline and DHM samples with a voltage of approximately 15–20 kV, which resulted in electric field (EF) strength between 108.5 to 123.0 kV/cm in this system. Due to the natural characteristic of the sample medium, the pulse width for the Milli-Q water was recorded at approximately 2.3 μs and EF strength at 93 kV/cm.

## Pulse waveform and energy calculation

A digital oscilloscope (InfiniiVision DSOX2024A; 200 MHz with 2 GSa/s, Keysight Technologies, Australia) was used to capture the instantaneous voltage and current waveforms using a high voltage probe (Tektronix P6015A, Tektronix, Inc., USA) and current monitor (Pearson wide-band current monitor, model 101, Pearson Electronics, USA). The energy, $E$ was calculated as per Equation 1 [19].

$$E = \int_{t_1}^{t_2} v(t)i(t)dt$$

(1)

Where, $E$ is the total energy of the pulse (Joule), $v(t)$ is the instantaneous voltage (V), and $i(t)$ is the current waveform (Ampere) delivered to the load from time $t_1$ to $t_2$.

The electric field strength for the parallel plate chamber is directly proportional to the applied voltage as per Equation 2 [19].

$$E_{field} = \frac{V}{d}$$

(2)

Where, $E_{field}$ is the electric field strength (kV/cm), $v$ is the applied voltage (kV), and $d$ is the distance between the parallel plate electrodes.

## Cleaning-in-place of the continuous PEF system

Between and after each test, tubes were replaced, and the entire system was cleaned using 1% w/v sodium hydroxide (NaOH; Chem-supply, Australia) circulated for 5 min. The system was then rinsed twice by circulating sterile distilled water for 5 min each at the maximum pump speed. To ensure the effectiveness of cleaning before each test, a sample was collected from the last recirculated sterile water and enumerated for the total microbial count using TSA.

## Conductivity measurement

The conductivity of different samples before PEF treatment was measured by a benchtop conductivity meter (Con 2700, Eutech Instruments Pte. Ltd., Singapore) equipped with a standard 4-cell conductivity probe (CONSEN9201D, Eutech Instruments, Singapore) that has the cell constant ($K_{cell}$) at 0.530 cm$^{-1}$. The cell constant is a geometric factor dependent on the probe used; it represents the ratio of the distance between the electrodes to their effective surface area within a conductivity cell. This setup was used to determine the specific conductance ($\kappa_{25\circ C}$) using Equations 3 and 4 [20]. The calibration was performed using a standard solution (1413 μS/cm @25°C, NIST Traceable solution, USA), and the probe was always rinsed with distilled water and dried before each analysis.

$$\kappa = K_{cell}G$$

(3)

$$\kappa_{25\circ C} = \kappa/(1 + \alpha(T - 25°C))$$

(4)

Where $G$ is the measured conductance at temperature, $T$, and $\alpha$ is the temperature-compensation factor (a value of 0.019 was used for $\alpha$). All samples were prepared at around ambient temperature, and the results were recorded as the mean of three measurements.

### Scanning electron microscope (SEM)

SEM was used to analyze the surface morphology changes in the bacteria in the sample pre- and post-PEF processing. The samples were centrifuged at 1500 rpm for 10 min to obtain the cell pellet and fixed using 2.5% glutaraldehyde in phosphate buffer saline (PBS) in a ratio of 10:1 at room temperature. The supernatant was removed, and the samples were washed twice in PBS. The samples were then applied to coverslips coated with poly-L-lysine (1 mg/mL) and left for 10 min for samples to adhere. Subsequently, the samples were dehydrated in a series of ethanol in a Pelco BioWave microwave (Ted Pella, Inc., Redding, USA) according to the manufacturer's instructions before being dried in a critical point dryer (Autosamdri-815 Series A, Tousimis, USA) according to the manufacturer's instructions. Coverslips were attached to stubs with double-sided carbon tabs and coated with platinum using a Compact Coating unit (CCU-010, Safematic, Switzerland) following the manufacturer's instructions. Imaging was carried out using a TM4000 tabletop SEM (Hitachi High-Tech, Japan) at an accelerating voltage of 10 kV and, for higher resolution imaging in a Sigma FE-SEM (Zeiss, Germany) at 2.25 kV.

### Transmission electron microscope (TEM)

The samples for TEM were prepared the same as those for SEM. For TEM, samples were enrobed in low-temperature gelling agarose (Sigma-Aldrich, USA), cut into 1 mm³ pieces, and post-fixed in 1% osmium tetroxide. The samples were then dehydrated in a series of ethanol, as per the SEM method, and infiltrated with Epon (Electron Microscopy Sciences, USA). Subsequently, the samples were polymerized in a 60°C oven for 48 h and sectioned at 80 nm with a diamond knife (Diatome, Switzerland) using a UC7 ultramicrotome (EM UC7, Leica Co., Ltd., Germany). Sections were placed on copper grids (ProSciTech, Australia) and contrasted with uranyl acetate and lead citrate. Imaging was carried out in an HT7700 TEM (Hitachi High-Tech, Japan) at an accelerating voltage of 80 kV.

### Statistical data analysis

For microbiological results, total count of surviving bacteria in CFU/mL were converted to $\log_{10}$ CFU/mL prior to statistical analysis. Count after processing below the limit of detection were assigned a value of 1 CFU/mL when calculating log reduction for statistical purposes as in literature [21]. All data presented in this study are means ± standard deviation (SD) from three independent experiments. The statistical analysis was performed using a two-way analysis of variance (ANOVA) following post hoc Tukey's test and one-way ANOVA with GraphPad Prism version 10.4.1, with a *p*-value lower than 0.05 considered a statistically significant difference.

## Results and discussion

### Voltage waveforms and conductivity measurement of saline solutions and donor human milk

To investigate the effect of PEF on HM while saving this precious sample, saline at various concentrations was first used as a substitute to optimise the system and processing conditions. Saline solutions at 0.05% to 0.2% salt concentrations were prepared and compared with HM samples to achieve similar conductivity and pulse waveforms. As recorded in Table 2, the specific electrical conductivity (mS/cm) of the Milli-Q water, 0.05%S, 0.1%S, 0.2%S, and HM samples were measured using the conductivity meter. Conductivity is a measurement of a solution's ability to conduct the electrical current [22]. Variation in the sample conductivity influences the current intensity required to generate EF and the distribution of EF during the PEF treatment [23,24]. Ensuring similar conductivity among samples also ensures similar applied voltage and pulse duration, which is further presented in Fig 2, as a single representative voltage waveform for the saline solution and HM samples over time. Pulse waveform refers to the shape and characteristics of the electrical pulses during PEF treatment. When two samples are subjected to similar pulse waveforms, it indicates consistent pulse shape, duration (width), polarity, and amplitudes between the samples [25]. This consistency further ensures that the samples receive similar EF strength and exposure time, leading to comparable effects of PEF treatment, such as microbial inactivation.

Table 2. Specific conductance measurements of all samples before PEF treatment. Data is represented as average±standard deviation (n=3).

| Sample | MQ | 0.05%S | 0.1%S | 0.2%S | HM |
|---|---|---|---|---|---|
| Average Specific conductance (mS/cm) ±STD | 0.002±0.000 | 1.156±0.003 | 2.221±0.004 | 4.505±0.014 | 2.300±0.018 |

As shown in Fig 2, all saline and HM samples exhibited a unipolar pulse shape with durations ranging from 68 to 128 ns (Table 1), whereas Milli-Q samples showed bipolar shapes indicating that conductivity affects pulse waveform. As previously mentioned, PEF treatment typically involves electric pulses lasting from milliseconds to microseconds. However, PEF treatment conducted in the nanoseconds range has been found to have negligible thermal effects while achieving similar membrane potentials with higher voltage differentials [26,27]. According to the literature, nsPEF has a relatively different mechanism for microbial inactivation than conventional PEF treatment. It can permeabilise the plasma cell membrane and induce injury accumulation in bacteria, leading to subsequent death. Since the pulses are shorter than the charging time of the outer membrane, energy from nsPEF focuses less on the lipid bilayer of the cellular membrane, but more on the intracellular material (e.g., mitochondria, nucleus, endoplasmic reticulum) compared to longer duration pulses (e.g., microsecond) [13,28–32]. Thus, nsPEF can inactivate bacteria more effectively while limiting the adverse impact on other components of the treated sample, as the effect is reversible. An additional and critical advantage of nsPEF treatment is that it can reduce electrode corrosion compared to conventional microsecond PEF treatment. The shorter pulse durations of nsPEF were shown to minimize the release of metal ions (e.g., aluminum) from electrodes, which is a side effect observed with conventional PEF treatment [33,34]. This reduction in corrosion not only enhances the safety and sustainability of the process but could also help to extend the longevity of the electrodes, which appears to be the major advantage of nsPEF over microsecond PEF treatment [35]. Although the specific energy input for nsPEF treatment in this study (Table 1) is comparable to the energy consumption reported for electricity-driven conventional thermal treatment systems (up to 511 kJ/L) [36], it is relatively higher than that of conventional PEF treatment. Therefore, for potential industrial application, the scalability of nsPEF will depend on optimising flow rate, electrode configuration, and pulse frequency to reduce total energy input while ensuring effective microbial inactivation. While the present study demonstrates effective inactivation at high EF strengths, further research is needed to determine the minimum effective field level required for reliable microbial safety. Additional cooling system, such as inline heat exchangers or continuous chilling loops might be necessary to prevent potential cumulative heating during processing of larger volumes. These factors highlight the need for further pilot development to confirm the nsPEF can deliver effective non-thermal pasteurisation while preserving heat-sensitive nutrients within feasible energy demands for milk processing.

Based on the results presented in Fig 2 and Table 2, 0.05%S exhibits a narrower pulse width (41 ns) and 0.2%S a wider pulse width (128 ns) as compared to HM samples, indicating a difference in energy exposure time. 0.1%S was found to be the closest match to the HM sample, with overlapping pulse waveform at a similar pulse width (76 ns vs.68 ns for 0.1%S and HM, respectively) and specific conductance (2.221 mS/cm vs. 2.300 mS/cm for 0.1%S and HM, respectively). The efficiency and predictability of the PEF treatment process are enhanced when both conductivity and pulse waveforms are consistent. Thus, using 0.1%S is beneficial for process optimisation, ensuring maximum effectiveness while minimizing variation. As expected, EF strength conducted at 0.1%S and HM were roughly the same at 117 and 116.4 kV/cm, respectively (Table 1).

## Effect of nsPEF treatment on microbiological inactivation in saline solutions

To first assess the efficacy of nsPEF treatment, experiments were conducted on saline samples with varying initial bacterial concentrations. PEF treatment using a high-energy pulsed generator was also applied to Milli-Q water samples with differential initial bacterial concentrations, as described in our previous work [18]. As presented in Table 3, two different

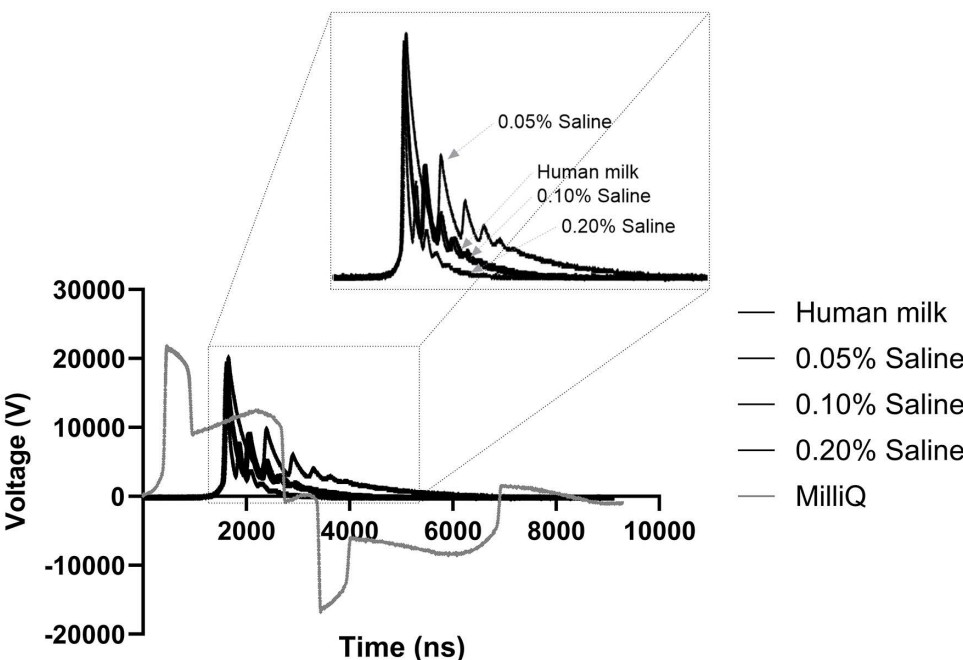

**Fig 2. Instantaneous voltage waveforms comparing MilliQ water, three different salts (NaCl) concentrations of saline (0.05%, 0.10%, and 0.20% w/v) and human milk in PEF treatment.**

**Table 3. Effectiveness of nsPEF treatment in *E. coli* inactivation (total bacterial count, log CFU/mL and log reduction) in inoculated 0.05% saline and 0.1% saline at various initial bacterial concentrations.**

| Sample | Initial bacterial count (log CFU/mL) | Bacterial count after nsPEF treatment (log CFU/mL) | Log reduction (CFU/mL) |
|---|---|---|---|
| 0.05%S | 5.4±0.4 | ND[1] | 5.4±0.4[2] (b) |
| | 7.4±0.3 | 1.8±0.4 | 5.6±0.3 (b) |
| | 8.4±0.2 | 3.3±0.8 | 5.1±0.6 (b) |
| 0.1%S | 5.4±0.4 | ND[1] | 5.4±0.3[2] (b) |
| | 7.4±0.3 | 1.9±0.4 | 5.6±0.4 (b) |
| | 8.4±0.2 | 1.5±0.4 | 6.9±0.4 (a) |

Values are presented as average±standard deviation (n = 3 independent experiments). Results within the log reduction column that do not share the same letters differ significantly ($p < 0.05$).

[1]ND = not detected; indicates bacterial count below the detection limit: ≤ 1 log CFU/mL

[2]As stated in the materials and methods, count after processing below the limit of detection were assigned a value of 1 CFU/mL when calculating log reduction for statistical purposes as in literature [21].

saline concentrations were selected as the closest matches to HM. These samples were inoculated with *E. coli* at 5, 7, and 8 log CFU/mL and subjected to nsPEF treatment under the same conditions. Nearly complete inactivation (based on the current detection limit) was achieved following nsPEF treatment across the two types of media with an initial count of 5 log CFU/mL. This result indicates the efficacy of using the nsPEF continuous system to achieve a microbial inactivation rate of 99.999%. Similar results were observed in the samples inoculated with 7 log CFU/mL of *E. coli*, where approximately 5 log reduction was achieved following the nsPEF treatment. For the initial count of 8 log CFU/mL, nearly 7 log

reduction was achieved in 0.1%S, while approximately 5.5 log reduction was recorded in 0.05%S. A two-way ANOVA showed a significant interaction between the two factors (sample*initial bacterial count) ($F_{(2,12)}$ = 9.736, $p = 0.0031$) and a significant sample effect ($F_{(1, 12)}$ = 10.83, $p = 0.0065$) on log reduction across the two samples. Tukey's post-hoc analysis confirmed that the 0.1% saline samples with an initial count of 8 log (Table 3) achieved significantly higher log reduction ($p < 0.05$). These results indicate that changing the medium conductivity from 1.156 mS/cm to 2.221 mS/cm (along with changing pulse waveform) does significantly affect the nsPEF treatment result under the current conditions. The results also indicate that the effect of initial concentration is likely to be medium dependent.

The effect of conductivity for PEF treatment in terms of microbial inactivation has been a controversial topic with different reported studies. Some studies have reported the highest microbial inactivation at the lowest conductivity [37,38]. Other studies have shown no difference in microbial inactivation when the same amount of total specific energy was applied [39,40]. Our previous study using a high-energy generator showed that a very small increase in specific conductance (from 3 to 40 µS/cm) in the treated samples enhanced the microbial inactivation effect when similar electric field strength and specific energy inputs were applied [18]. Thus, it is generally difficult to analyze PEF literature on conductivity or pulse waveform as the processing parameters were not consistent across different studies due to the use of different pulsed power generators, liquid samples, cuvette, and treatment chamber dimensions. These factors can influence the electrical properties of the system, which consequently affects the pulse waveform. Additionally, conductivity is not a constant factor but changes with temperature, and processed food is generally not a homogenous system, which means different conductivity levels might be found in various parts of the samples [41].

## Effect of nsPEF treatment on microbiological inactivation in inoculated pasteurised human milk

The saline solution results were sufficient to demonstrate the efficacy of nsPEF treatment, with more than 5 log reduction recorded at various initial concentrations as a common requirement for HM processing, which is typically achieved by heat pasteurisation [42]. Consequently, HM samples were also inoculated with various concentrations of *E. coli* and treated with nsPEF at the same condition, as real-world samples often vary in their bacteria concentration, and this variability could significantly affect the nsPEF treatment efficacy. Although comparable electrical properties (voltage waveforms and conductivity) were found between 0.1%S and HM samples, the HM samples exhibited distinct results. As presented in Table 4, approximately 3 log reduction was achieved following nsPEF treatment of inoculated human milk with an initial count of about 5 log CFU/mL, while nearly 5 log reduction was recorded with an initial count of 7 and 8 log CFU/mL, respectively. It is noteworthy that a consistent residual count of approximately 2–3 log of *E. coli* persisted across all HM samples post-nsPEF treatment, irrespective of the initial bacteria count. Additionally, one-way ANOVA with Tukey's post-hoc analysis for HM samples (Table 4) showed that sample with an initial count of 5 log was significantly different from the other two ($p < 0.05$). This further supports the statement that the effect of initial bacterial concentration is medium dependent, as observed in the 0.1% saline samples (Table 3).

**Table 4. Effect of nsPEF treatment in *E. coli* inactivation (total bacterial count, log CFU/mL and log reduction) in inoculated donor human milk samples at various initial bacterial concentrations.**

| Sample | Initial bacteria count (log CFU/mL) | Bacteria count after nsPEF treatment (log CFU/mL) | Log reduction (CFU/mL) |
|---|---|---|---|
| HM | 5.6 ± 0.3 | 2.2 ± 0.1 | 5.4 ± 0.4 (b) |
| | 7.3 ± 0.3 | 2.8 ± 0.4 | 5.6 ± 0.3 (a) |
| | 8.4 ± 0.3 | 3.4 ± 0.5 | 5.1 ± 0.6 (a) |

Values are present as average ± standard deviation (n = 3 independent experiments). Results within the log reduction column that do not share the same letters differ significantly ($p < 0.05$).

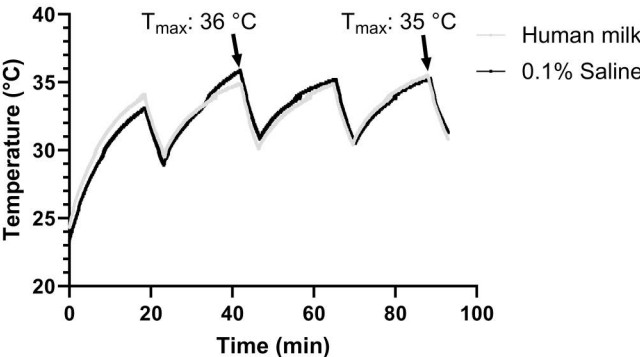

**Fig 3. Temperature monitoring over time for 0.1% saline and human milk samples during nsPEF treatment using an FO sensor.**

## Temperature measurement

Temperature control is a critical aspect of non-thermal pasteurisation, especially when treating a medium with relatively higher conductivity and using treatment with higher energy expenditure. In this study, temperature measurement was conducted using both an FO sensor and an infrared thermal camera. Fig 3 below illustrates the temperature changes during the entire nsPEF treatment in various steps, with the aid of the ice pack jacketed around the sample reservoir. It can be observed that the temperature drops after each pulse step during the resting period and rises again during the treatment. The maximum temperature for the 0.1%S and HM samples was recorded at 36°C and 35°C, respectively, with the initial temperature at approximately 24°C. Similar results were obtained using the infrared thermal camera, with the maximum temperature recorded at the end of each step being below 35°C. However, although the maximum temperature observed during nsPEF treatment was far below the conventional pasteurisation thresholds (62.5°C), this moderate increase (~11°C) could potentially influence the quality of human milk. Future work should investigate possible effects on orogenetic and nutritional properties to fully validate nsPEF as a particle alternative.

## SEM

To further confirm the bacterial inactivation effect and evaluate the morphological changes of the bacteria in the samples after nsPEF treatment, SEM analysis was conducted on pre- and post-treated saline and HM samples inoculated with an initial count of 8 log CFU/mL *E. coli*. As presented in Fig 4a, intact *E. coli* cells in rod shape were observed before nsPEF treatment, while cell wall disruption was observed in the treated samples (Fig 4b). A similar observation was also reported in phosphate buffer saline [43], but the authors noted that the disruption was mainly at the poles of the bacterial rods after PEF treatment, while cell wall damage was observed only after plasma treatment in their study. In this study, different results were observed, with damage found on both the pole and the sides of the bacteria, similar to the recent study in saline solution [44]. This result could be due to the difference in the PEF system settings or variations in the process parameters, as a continuous system combined with a magnetic stirrer was used, which further enhanced the uniformity of the nsPEF treatment. Most of the cells present in Fig 4b show varying levels of cell wall damage due to electroporation, compromising their integrity and potentially leading to cell death. This result corresponds to the microbial inactivation results in Table 3, where only 1.5 log CFU/mL remained after nsPEF treatment in a sample with an initial count of 8 log CFU/mL.

The SEM images of HM samples were distinct from those of saline samples both before and after nsPEF treatment. Obvious debris was found around the *E. coli* cell before the nsPEF treatment, and most of the cells were clumped together around the debris. The debris is likely the result of pasteurisation effects on human milk fat globules (HMFG)

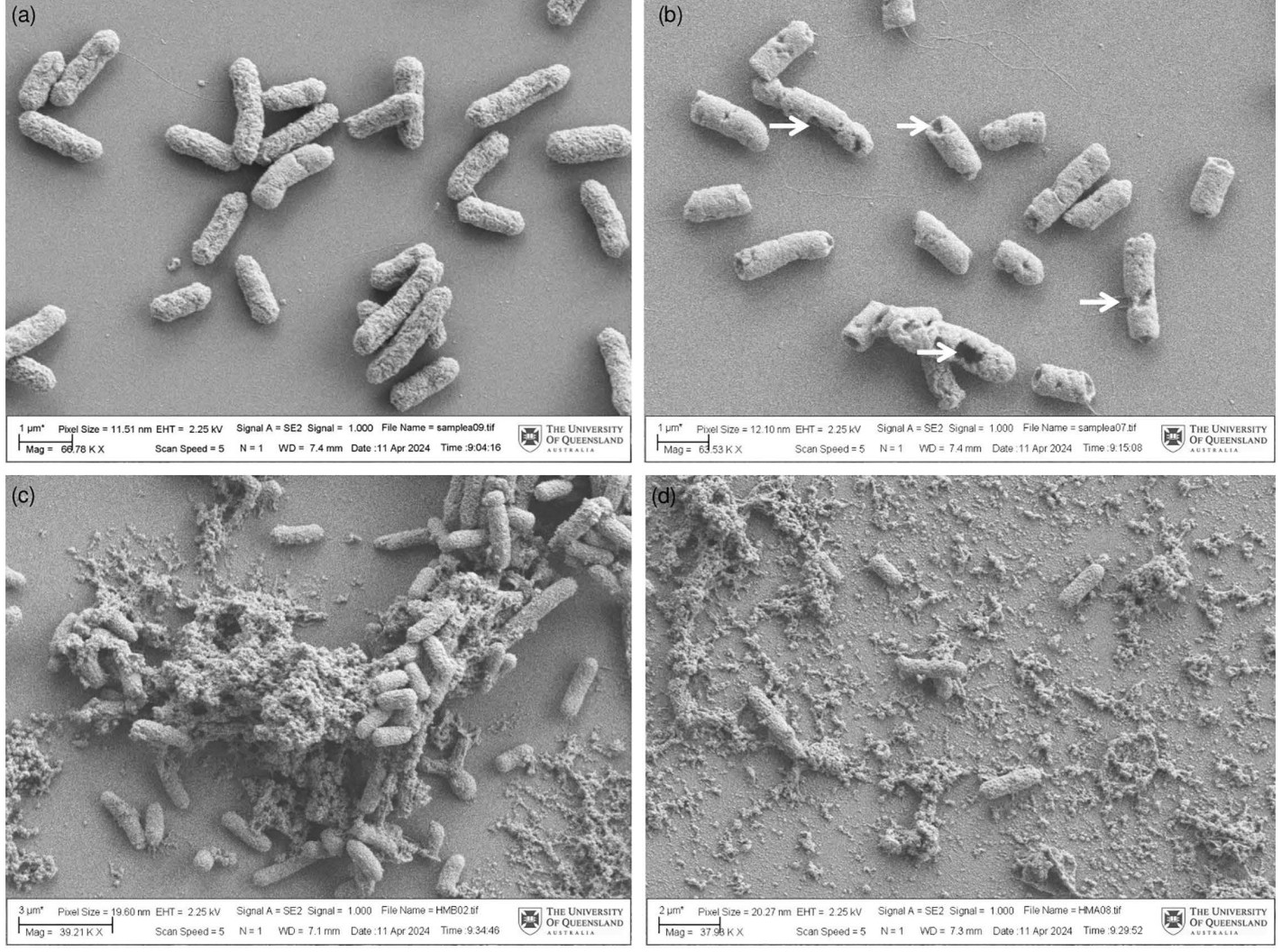

**Fig 4. Scanning Electron Microscopy of (a) 0.1% saline samples with 8 logs inoculated *E. coli* before nsPEF treatment at 66780X magnification and (b) after nsPEF treatment at 63530X magnification; (c) pasteurised human milk samples with 8 logs inoculated *E. coli* before nsPEF treatment at 39210X magnification and (d) after nsPEF treatment at 37980X magnification.** The white arrow indicates some of the cell disruption areas.

before inoculation and epithelial cells present in HM, as reported by other studies [45,46]. However, after the nsPEF treatment, the number of colonies was significantly reduced at a similar magnification with dispersive debris around them, and no clear damage to the cell wall was observed on these cells. The debris might have a protective effect on the *E. coli*, interrupting the microbial inactivation effect of nsPEF treatment in HM. Previous electric field modelling studies by Toepfl, Heinz [47] demonstrated that agglomeration of microbial cells or their interaction with insulating particles (e.g., fat globules) can reduce the lethality of PEF treatment. Using 2D electric field modelling, the researchers showed that such agglomerations disrupt local electric field distribution, preventing the membrane from achieving the critical transmembrane potential required for electroporation. The findings highlight the importance of selecting an appropriate electric field strength that exceeds the critical threshold for all cells, particularly in complex food matrices such as human milk.

## TEM

TEM was conducted to further investigate the structure of the bacteria in nsPEF-treated HM samples. Three clear and intact layers (outer membrane, peptidoglycan-made cell wall, and cytoplasmic membrane) were observed in Fig 5a pre-nsPEF samples as the characteristics of gram-negative bacteria [48]. Similar to the phosphate buffer study [49], Fig 5b shows that *E. coli* treated with nsPEF appeared to have ruptured cell walls and leakage of the cytoplasmic material, which could further lead to changes in the intracellular organisation. However, not all bacteria were subject to similar damage. Based on Fig 5c, although cells with ruptured cell walls and enlarged vacuoles were found, there were also cells with intact cell walls and no noticeable alteration. This might be due to the presence of HMFG and epithelial cells in HM, which could potentially protect the cells from nsPEF treatment (as seen in Fig 4c, d). This potential protecting effect might also explain why there were consistently 2–3 log of bacteria remaining after nsPEF treatment, irrespective of the initial count. However, further analysis would be needed to investigate these milk components to draw a definitive conclusion.

## Conclusions

This study investigated the efficacy of nsPEF treatment as a non-thermal pasteurisation method for donor human milk. While complete bacterial inactivation was achieved in the model saline solution with an initial *E. coli* count of 5 log CFU/mL, reduced efficacy was observed in human milk samples. There were always around 2–3 log bacteria remaining after nsPEF treatment in the human milk samples, irrespective of the initial count. This result indicates that the matching pulse waveforms and electrical conductivity between saline and human milk samples do not guarantee similar bacterial inactivation effects, and sample composition plays a significant role in determining process efficacy. Microscopy analysis results suggested that components like human milk fat globules and epithelial cells present around the *E. coli* cells in the human milk samples likely provide protective effects for the bacteria and influence the efficacy of nsPEF-induced inactivation. Although nsPEF showed some limitations in microbial inactivation in human milk samples, the treatment was carried out at significantly lower temperatures than conventional thermal pasteurisation, highlighting the potential of nsPEF treatment in preserving the nutritional and immunological quality of donor human milk. Overall, nsPEF treatment holds the potential to be an alternative pasteurisation method for donor human milk. Future work should focus on optimising the treatment

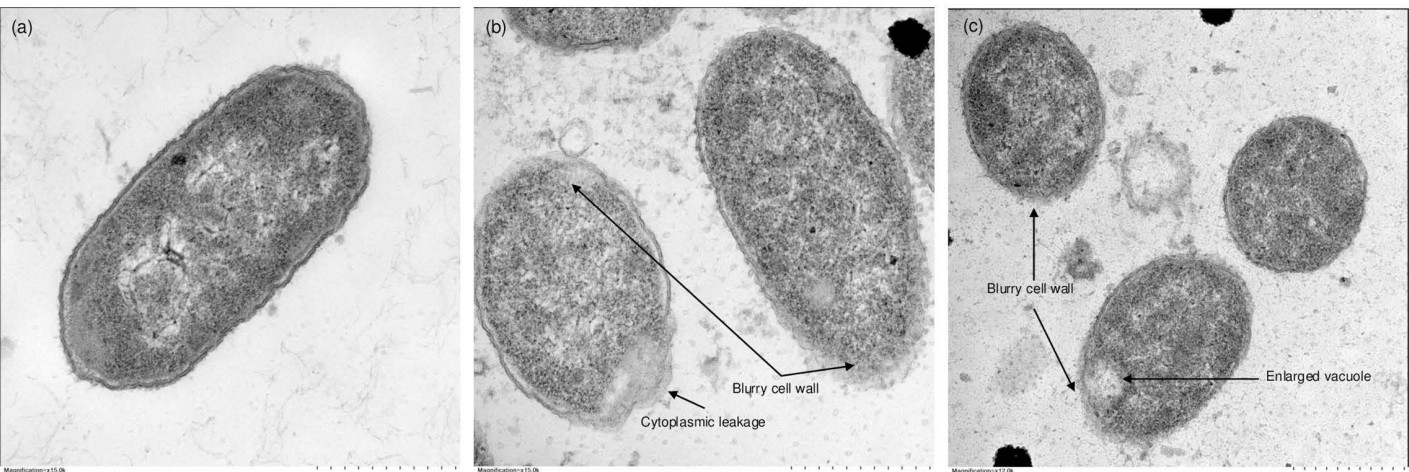

**Fig 5. Transmission Electron Microscopy of (a) pasteurised human milk samples with 8 log CFU/mL inoculated *E. coli* before nsPEF treatment at 15000X magnification, (b) after nsPEF treatment at 15000X magnification and (c) after nsPEF treatment at 12000X magnification.**

parameters to overcome the matrix-related shielding effects, consider the scalability for potential industrial application, and comprehensively evaluating the impact of nsPEF on the physicochemical and nutritional properties of human milk.

## Acknowledgments

The authors would like to thank the generous mothers who donated their milk and the staff at the Royal Brisbane Women's Hospital who helped in organising the milk samples. The authors acknowledge the facilities, and the scientific and technical assistance, of the Australian Microscopy & Microanalysis Research Facility at the Centre for Microscopy and Microanalysis, The University of Queensland.

## Author contributions

**Conceptualization:** Yiting Wang, Farzan Zare, Negareh Ghasemi, Firuz Zare, Nidhi Bansal.

**Data curation:** Yiting Wang.

**Formal analysis:** Yiting Wang, Buddhi Dayananda.

**Funding acquisition:** Negareh Ghasemi, Firuz Zare, Paul N. Shaw, Pieter Koorts, Nidhi Bansal.

**Investigation:** Yiting Wang, Farzan Zare, Elisabeth K. Prabawati.

**Methodology:** Yiting Wang, Farzan Zare, Elisabeth K. Prabawati, Negareh Ghasemi, Firuz Zare, Nidhi Bansal.

**Project administration:** Nidhi Bansal.

**Resources:** Pieter Koorts.

**Supervision:** Negareh Ghasemi, Firuz Zare, Paul N. Shaw, Nidhi Bansal.

**Visualization:** Yiting Wang, Farzan Zare.

**Writing – original draft:** Yiting Wang, Farzan Zare.

**Writing – review & editing:** Elisabeth K. Prabawati, Buddhi Dayananda, Negareh Ghasemi, Firuz Zare, Paul N. Shaw, Pieter Koorts, Nidhi Bansal.

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
