## [Decision Letter · Decision Letter 0]

27 May 2025

PONE-D-25-10160
Evaluating continuous nanosecond pulsed electric field (nsPEF) treatment as a non-thermal alternative for human milk pasteurisation
PLOS ONE

Dear Dr. Bansal,

Thank you for submitting your manuscript to PLOS ONE. After careful consideration, we feel that it has merit but does not fully meet PLOS ONE’s publication criteria as it currently stands. Therefore, we invite you to submit a revised version of the manuscript that addresses the points raised during the review process.

We look forward to receiving your revised manuscript.

Kind regards,

Olga Zeni

Academic Editor

PLOS ONE

Journal Requirements:

 “Children’s Hospital Foundation Innovator Grant (50317) and the National Health and Medical Research Council Ideas Grant (GNT1182038).” 

3. We note that your Data Availability Statement is currently as follows: All relevant data are within the manuscript and in Supporting Information files.

Reviewers' comments:

Reviewer's Responses to Questions

**Comments to the Author**

1. Is the manuscript technically sound, and do the data support the conclusions?

Reviewer #1: Partly

Reviewer #2: No

2. Has the statistical analysis been performed appropriately and rigorously? 

Reviewer #1: Yes

Reviewer #2: No

3. Have the authors made all data underlying the findings in their manuscript fully available?

Reviewer #1: Yes

Reviewer #2: Yes

4. Is the manuscript presented in an intelligible fashion and written in standard English?

Reviewer #1: Yes

Reviewer #2: Yes

5. Review Comments to the Author

Reviewer #1: The authors provide a study on nsPEF treatment of human donor milk aiming upon microbial inactivation to enhance product safety. The paper is well written and of high scientific as well as societal interest. ns is a promising technique.

I would recommend to add further literature on use of PEF and nsPEF in liquid food treatment, in particular dairy or other protein rich products such as liquid egg or blood.

A. Shivani Indumathi, G. Sujatha, V. Appa Rao, Rita Narayanan, C.N. Kamalarathnam, V. Perasiriyan A. and Serma Saravana Pandian (2022). Effect of Pulsed Electric Field on Physicochemical Parameters and Nutrient Content of Mother’s Milk. Biological Forum– An International Journal, 14(2a): 572-575

S, S., G, S., Narayanan, R., & V, P. (2022). STORAGE STUDIES ON PULSED ELECTRIC FIELD PROCESSED MOTHER’S MILK. Indian Journal of Veterinary and Animal Sciences Research, 50(5), 76-80.

nsPEF is claimed to require less energy / cause less product damage than PEF. That statement needs further justication, in particular as the energy input levels reported in the study are very high. With energy input levels of 500 to 600 kJ/l commercial scalability is questionable, and also significant heating is to be expected. Even if under the conditions used in the lab trials (small treatment volume / large electrode area / long treatment time / time to cool down) the peak temperatures were low, when scaling up time behavior will be different and more heating occur. Those effects, potential limitations or ways to handle should be discussed.

How much cool-down do you expect during the transport from the treatment chamber to the reservoir? Which temperature peaks are expected during the treatment, can you calculate those using the residence time in the chamber and the energy input?

It is mentioned that nsPEF leads to less corrosion. Has any corrosion been observed using 316 stainless steel electrodes, and would you expect undesired effects on HM composition?

The conclusion state that molk fat gobules might have a protective effect. Such effects have been described by field modelling, e.g. in https://doi.org/10.1016/j.cep.2006.07.011

Very high field strength levels have been used. Did you perform a study if those are required or are they more or less a result of available / resulting treatment conditions. Minimum required field levels would be of high interest for equipment design.

I would recommend to revise and discuss above points prior to publication.

Reviewer #2: In this paper, nanosecond pulsed electric fields (nsPEFs) were tested as an alternative, non thermal technique for the pasteurization of donated human milk. The topic of the paper is potentially interesting, however, the adopted methodologies present major issues that make the paper not recommendable for publication in this reviewer's opinion.

Detailed comments are following.

Major issues

1) LInes 134-135: "a total of 240,000 pulses (60,000 pulses per step, 5 min 135 resting time between each step) were applied at 50 Hz frequency using a pulsed power generator as the optimised processing condition." Are these pulse numbers consistent with the flow rate and the pulse repetition time?

2) Figure 1 and sectionn 2.3: Please, provide more details regarding the pulse generator. I understand it is a commercial pulse generator, but I think it would be useful to describe the circuit topolgy since it covers quite a large range of pulse durations (from 2 us down to 40 ns).

3) Figure 1 and section 2.3: Details on the geometric characteristics of the cuvette should be provided. I guess it is based on parallel plate electrodes. However, as I understand from the description and by the figure, the exposed medium is flowing through a plastic tube. This implies that pulses were not actually delivered through conductive coupling, but through capactivie coupling instead, which would imply a reduction of the induced electric field with respect to the applied one.

4) Table 1: How were the E-field values assesssed? Is it the voltage-to-distance ratio considering the voltage applied on the electrodes? If so, based on the previous comment, this should be replaced by a more accurate calculation considering that the media were exposed while flowing in a tube, and not in direct contact with the electrodes.

5) Section 2.4 (Pulse waveform and energy measurement): The energy was not measured. It was calculated. Anyway, this parameter was not used in the rest of the paper.

6) Line 172: Please, provide a definition/physical meaning for this "cell constant"

7) Lines 265-267: As I understand, the authors tested saline solutions with different concentrations to identify the one having a conductiviy level similar to that of human milk. If so, this should be better explained at the beginning of section 3.1

What I don't catch is the reason why the different media are exposed to different pulse durations. The different conductivity values affect the current and, hence, the induced electric field, but should not affect the pulse durations, unless I am missing something regarding the pulse generation modality, which gets back to my previous comment regarding clarifications on the pulse generator.

8) Lines 293-298: The statistically significant differences should be included in Table 3. Also, I guess the subscript numbers in the statistics refer to the different experimental conditions, but these have not been defined.

9)LInes 309 - 312: If the manuscript has not been published yet, it should not be cited

10) Figure 2 (Pulse shape) : The bipolar aspect of pulses in the case of MilliQ could be due to the capacitive coupling situation, rather than to the different conductivity. It is strange, however, that the other pulses are not bipolar as well. Moreover, nsPEFs are supposed to be rectangular, which are not in the figure shown here, and also in some conditions they present spurious pulses with peak values that are comparable to the main one. This is another aspect that recalls the pulse generation issue. Please, elaborate on that.

11) Figure 3 (Temperature): I understand that the heating effect under nsPEFs was not as much as that in conventional heating pasteurization techniques. However, tha temperature excursion is not negligible (almost 10 °C). Can this be called a "non-thermal effect"? Could this temperature excursion affect the organoleptic properties of the milk? Please, comment on that.

Minor issues

1) Introduction section, line 65. "In the previous study,..." Do you mean "In our previous study"?

2) Lines 133-134. "Based on the previous studies with Milli-Q water (manuscript 134 submitted and yet to be published)..." If the manuscript has not been published yet, then it should not be mentioned.

6. PLOS authors have the option to publish the peer review history of their article (what does this mean?). If published, this will include your full peer review and any attached files.

Reviewer #1: **Yes: **Stefan Toepfl

Reviewer #2: No

---

## [Author Response · Author response to Decision Letter 1]

11 Jul 2025

Dear Editor,

Thank you again for your time and consideration of our submission.

1. We have revised the manuscript to ensure it fully aligns with PLOS ONE’s style requirements, including those for file naming.

2. For the financial disclosure, we would like to state that “The funders had no role in study design, data collection and analysis, decision to publish, or preparation of the manuscript”.

3. Regarding the minimal data set, we have included all relevant raw data in a repository with corresponding URL. The DOI will be available only after acceptance of the manuscript for publication so that we can ensure their inclusion before publication.

Dear Reviewers,

Thank you once again for your time and thoughtful comments on our manuscript. Please find our detailed point-by-point responses in the attached word document [Response to Reviewers] and below.

Reviewer #1:

The authors provide a study on nsPEF treatment of human donor milk aiming upon microbial inactivation to enhance product safety. The paper is well written and of high scientific as well as societal interest. ns is a promising technique. I would recommend to add further literature on use of PEF and nsPEF in liquid food treatment, in particular dairy or other protein rich products such as liquid egg or blood. A. Shivani Indumathi, G. Sujatha, V. Appa Rao, Rita Narayanan, C.N. Kamalarathnam, V. Perasiriyan A. and Serma Saravana Pandian (2022). Effect of Pulsed Electric Field on Physicochemical Parameters and Nutrient Content of Mother’s Milk. Biological Forum– An International Journal, 14(2a): 572-575S, S., G, S., Narayanan, R., & V, P. (2022). STORAGE STUDIES ON PULSED ELECTRIC FIELD PROCESSED MOTHER’S MILK. Indian Journal of Veterinary and Animal Sciences Research, 50(5), 76-80.

Author response and action:

We thank the reviewer for the insightful comments and helpful references. We agree that including additional references on the use of PEF or nsPEF in liquid food, particularly in human milk, would be beneficial for the article. We have therefore updated the references in the introduction section from line 72 to 77 and as below:

“A study by Sanchaya and Sujatha (14) compared raw DHM to PEF-treated DHM samples and reported effective microbial reduction with no significant changes in pH, acidity, or fat content. Similarly, Indumathi and Sujatha (15) applied PEF in DHM and also reported no significant change in pH, acidity, fat, lactose, and ascorbic acid content between PEF-treated and raw DHM. However, a significant reduction in protein was observed, which the authors attributed to potential deposition of milk solids on the electrode surfaces."

nsPEF is claimed to require less energy / cause less product damage than PEF. That statement needs further justication, in particular as the energy input levels reported in the study are very high. With energy input levels of 500 to 600 kJ/l commercial scalability is questionable, and also significant heating is to be expected. Even if under the conditions used in the lab trials (small treatment volume / large electrode area / long treatment time / time to cool down) the peak temperatures were low, when scaling up time behavior will be different and more heating occur. Those effects, potential limitations or ways to handle should be discussed.

Author response and action:

We thank the reviewer for highlighting this important point. We agree that the specific energy input reported in this study are relatively high, which partly result from the current experimental setup optimised for proof-of-concept rather than industrial scale. We have revised and added a new paragraph in the Result and Discussion section acknowledging that while nsPEF pulses have been shown to have reduced local thermal effect due to a relatively different mechanism for microbial inactivation than conventional PEF treatment, the total energy input at lab scale can still lead to some level of heating. We also discuss that practical scale-up would require design improvements, such as optimised flow-through chambers, reduced processing time, enhanced heat exchange, and more efficient energy delivery systems to ensure feasible energy efficiency and maintain the non-thermal advantage. The relevant paragraph has been added to line 283 to 288 and 290 to 295, and is shown below:

“Although the specific energy input for nsPEF treatment in this study (Table 1) is comparable to the energy consumption reported for electricity-driven conventional thermal treatment systems (up to 511 kJ/L) [34], it is relatively higher than that of conventional PEF treatment. Therefore, for potential industrial application, the scalability of nsPEF will depend on optimising flow rate, electrode configuration, and pulse frequency to reduce total energy input while ensuring effective microbial inactivation. … Additional cooling system, such as inline heat exchangers or continuous chilling loops might be necessary to prevent potential cumulative heating during processing of larger volumes. These factors highlight the need for further pilot development to confirm the nsPEF can deliver effective non-thermal pasteurisation while preserving heat-sensitive nutrients within feasible energy demands for milk processing.”

How much cool-down do you expect during the transport from the treatment chamber to the reservoir? Which temperature peaks are expected during the treatment, can you calculate those using the residence time in the chamber and the energy input?

Author response and action:

We would like to thank the reviewer for the insightful comment. Yes, this is possible using the following equation (We have approximated fluid density in the traditional equation to mass as in this example we are using water):

∆T=Q/(mc_p )=Pt/(mc_p )

cp for water assumed to be roughly 4.2 J/g°C

Mass of water within the treatment volume of (40 mm x 2.65 mm x 1.65 mm) = 0.1749 g

The flowrate is set to 100 mL/min or 1.67 mL/s, therefore the residence time is 0.105 seconds assuming a volume of 0.1749 mL. We can exclude the influence of the Venturi effect if we assume that the peristaltic pump is keeping a constant flowrate throughout the closed-loop system.

Since our pulses are being applied at a 50 Hz frequency, we have 5.25 pulses (50×0.105) applied to the liquid water during the residence time of 0.105 seconds. So, the total energy input during the residence time is 13.125 J. Therefore, the total theoretical temperature increase is roughly 18 °C during the 0.105 s residence time. In other words, peak theoretical outlet temperature is expected to be 42 °C.

Comparatively with the results presented in the manuscript (Figure 3), given an initial temperature of approximately 24 °C with a final temperature of 36 °C (measured at the reservoir), the difference or rise in temperature is estimated to be 12 °C. Therefore, we can very roughly estimate the cooling between the treatment chamber and the reservoir to be 6 °C.

It is mentioned that nsPEF leads to less corrosion. Has any corrosion been observed using 316 stainless steel electrodes, and would you expect undesired effects on HM composition?

Author response and action:

We thank the reviewer for pointing this out. In our current study, no obvious corrosion was observed when using 316 stainless steel electrodes, which are commonly used in industrial applications and are considered a food-grade material. Therefore, we do not expect to see any undesired effects on human milk under current test conditions. However, in our preliminary tests using cuvettes made of aluminium, we did observe visible corrosions (Data not included in this current study).

The conclusion state that milk fat globules might have a protective effect. Such effects have been described by field modelling, e.g. in https://doi.org/10.1016/j.cep.2006.07.011

Author response and action:

We thank the reviewer for providing this very helpful reference, we have revised the Result and Discussion (line 437-444) and Conclusion section (line 476, 482-485) accordingly.

The added paragraph in the Result and Discussion section is provided below:

“Previous electric field modelling studies by Toepfl, Heinz (45) demonstrated that agglomeration of microbial cells or their interaction with insulating particles (e.g., fat globules) can reduce the lethality of PEF treatment. Using 2D electric field modelling, the researchers showed that such agglomerations disrupt local electric field distribution, preventing the membrane from achieving the critical transmembrane potential required for electroporation. The findings highlight the importance of selecting an appropriate electric field strength that exceeds the critical threshold for all cells, particularly in complex food matrices such as human milk.”

The added paragraph in the Conclusion section is also provided below:

“Future work should focus on optimising the treatment parameters to overcome the matrix-related shielding effects, consider the scalability for potential industrial application, and comprehensively evaluating the impact of nsPEF on the physicochemical and nutritional properties of human milk.”

Very high field strength levels have been used. Did you perform a study if those are required or are they more or less a result of available / resulting treatment conditions. Minimum required field levels would be of high interest for equipment design.

Author response and action:

We appreciate the reviewer for this valuable point. We acknowledge that the electric field strength used in this study are relatively higher compared to conventional microsecond PEF treatments, which is a recognised characteristic of nsPEF. These values in the current study were mainly determined by the available pulse generator and the lab-scale continuous chamber set up optimised for proof-of-concept nsPEF application. We did not conduct an extensive parameter trials to determine the absolute minimum required electric field strength for effective microbial inactivation in human milk, as the primary aim was to demonstrate feasibility under defined conditions. However, we agree that identifying the minimum effective electric field strength would be highly relevant for future equipment design and energy optimisation. We have added a statement in the Result and Discussion section (line 288 - 290) to highlight this point and recommended that future scale up work should focus on optimise field strength while maintaining efficacy and minimising energy demand.

The added paragraph in the Result and Discussion section is also provided below:

“While the present study demonstrates effective inactivation at high EF strengths, further research is needed to determine the minimum effective field level required for reliable microbial safety.”

Reviewer #2:

In this paper, nanosecond pulsed electric fields (nsPEFs) were tested as an alternative, non thermal technique for the pasteurization of donated human milk. The topic of the paper is potentially interesting, however, the adopted methodologies present major issues that make the paper not recommendable for publication in this reviewer's opinion.

Detailed comments are following.

Lines 134-135: "a total of 240,000 pulses (60,000 pulses per step, 5 min 135 resting time between each step) were applied at 50 Hz frequency using a pulsed power generator as the optimised processing condition." Are these pulse numbers consistent with the flow rate and the pulse repetition time?

Author response and action:

We appreciate the reviewers’ comment and the chance to address this point. A total of 240,000 pulses were applied to the 100 mL sample volume, delivered in four steps of 60,000 pulses each. The frequency determines the total treatment time but does not affect the flow rate. Meaning that with 60,000 pulses at 50 Hz we have a testing time of 60,000x0.02s=1200s or 20 minutes. As shown in Figure 3, we have demonstrated that the temperature of the sample under the test was kept below 36°C, when we applied 60,000 pulses with resting time instead of continuous pulse excitation (240,000) which could increase the temperature of the sample higher than 36°C. The flowrate is controlled independently by the peristaltic pump used, which in our case was 100 mL/min. Therefore, the flowrate and the pulse number are independent variables and do not affect each other.

Figure 1 and section 2.3: Please, provide more details regarding the pulse generator. I understand it is a commercial pulse generator, but I think it would be useful to describe the circuit topolgy since it covers quite a large range of pulse durations (from 2 us down to 40 ns).

Author response and action:

We thank the reviewer for this comment and appreciate the opportunity to clarify. The pulse generator used in this study is designed for high-current, high-power applications typically depends on the system's load characteristics, and as a result, the pulse duration is not easily adjustable. These systems are often designed around specific load conditions, limiting operational flexibility. In contrast, pulsed power systems for low-current applications—such as dielectric barrier discharge (DBD)—can utilize hard-switching semiconductor devices commonly used in power electronics. These enable precise control of pulse frequency, amplitude, and duration. Our current pulsed generator, designed for high-power operation, features a fixed pulse duration due to its load-driven nature, unlike power-electronics-based systems used in low-power, tunable applications like DBD. Therefore, the pulse generator used does not allow for a fixed, selectable range of pulse durations from 2 μs down to 40 ns as describe in the comment. The pulse generator is a dependent voltage source meaning that it is load-dependent. The output voltage, pulse shape, and pulse duration are set by the load’s impedance. Meaning that if the pulse generator is applied to a high-impedance load the output voltage, pulse shape, and pulse duration will be different than when applied to a low-impedance load.

Figure 1 and section 2.3: Details on the geometric characteristics of the cuvette should be provided. I guess it is based on parallel plate electrodes. However, as I understand from the description and by the figure, the exposed medium is flowing through a plastic tube. This implies that pulses were not actually delivered through conductive coupling, but through capactivie coupling instead, which would imply a reduction of the induced electric field with respect to the applied one.

Author response and action:

We thank the reviewer for this comment and the opportunity to clarify. The measurement of the parallel plate electrode geometry is provided in the manuscript on line 139-140 “(L x W x D: 40 mm x 2.65 mm x 1.65 mm)”. The exposed medium is in direct contact with the electrodes in the cuvette (marked “4” in Figure 1) and not inside the plastic tubing. The plastic tubing is used only to transfer the liquid continuously between the cuvette, pump, and reservoir. We have added a brief clarification in the Figure 1 caption to emphasise this point and to avoid any misunderstanding about conductive coupling as “The liquid sample flows directly between the parallel plate electrodes in the cuvette and is not enclosed within the plastic tubing during exposure”. Additional detail on the cuvette geometry can also be found in the updated reference in the manuscript (reference #18). We appreciate the reviewer’s observation, which helped us ensure this aspect is described more clearly.

Table 1: How were the E-field values assessed? Is it the voltage-to-distance ratio considering the voltage applied on the electrodes? If so, based on the previous comment, this should be replaced by a more accurate calculation considering that the media were exposed while flowing in a tube, and not in direct contact with the electrodes.

Author response and action:

We thank the reviewer for this comment. The electric field values were calculated using the standard parallel-plate electrode equation:

Efield= V/d

Where, Efield is the electric field strength (kV/cm), V is the applied voltage (kV), and d is distance between the parallel plate electrodes.

As clarified in our response to comment#3 above, the PEF treatment is applied directly

---

## [Decision Letter · Decision Letter 1]

24 Sep 2025

Evaluating continuous nanosecond pulsed electric field (nsPEF) treatment as a non-thermal alternative for human milk pasteurisation

PONE-D-25-10160R1

Dear Dr. Nidhi Bansal, 

We’re pleased to inform you that your manuscript has been judged scientifically suitable for publication and will be formally accepted for publication once it meets all outstanding technical requirements.

Kind regards,

Olga Zeni

Academic Editor

PLOS ONE

Reviewer #2: All comments have been addressed

2. Is the manuscript technically sound, and do the data support the conclusions?

Reviewer #2: Yes

3. Has the statistical analysis been performed appropriately and rigorously? 

Reviewer #2: Yes

4. Have the authors made all data underlying the findings in their manuscript fully available?

Reviewer #2: Yes

5. Is the manuscript presented in an intelligible fashion and written in standard English?

Reviewer #2: Yes

6. Review Comments to the Author

Reviewer #2: The authors have adequately addressed the issues raised in the first revision round and the paper is now acceptable for publication in this reviewer's opinion

7. PLOS authors have the option to publish the peer review history of their article (what does this mean?). If published, this will include your full peer review and any attached files.

Reviewer #2: No

---

## [Editor Report · Acceptance letter]

PONE-D-25-10160R1

PLOS ONE

Dear Dr. Bansal,

I'm pleased to inform you that your manuscript has been deemed suitable for publication in PLOS ONE. Congratulations! Your manuscript is now being handed over to our production team.

Kind regards,

on behalf of

Dr. Olga Zeni

Academic Editor

PLOS ONE